

# Seasonal variation and chemical characterization of PM$_{2.5}$ in northwestern Philippines

Gerry Bagtasa[1], Mylene G. Cayetano[1], and Chung-Shin Yuan[2]

[1]Institute of Environmental Science & Meteorology, University of the Philippines, Diliman, Quezon City, Philippines
[2]Institute of Environmental Engineering, National Sun-Yat Sen University, Kaoshiung, Taiwan

*Correspondence to:* Gerry Bagtasa (gbagtasa@iesm.upd.edu.ph)

**Abstract.** The seasonal and chemical characteristic of fine particulate matter (PM$_{2.5}$) was investigated in Burgos, Ilocos Norte, located at the northwestern edge of the Philippines. Each 24H-sample of fine aerosol was collected for two weeks every season. Fine particulate in the region shows strong seasonal variation in both concentration and composition. Highest mass concentration was seen during the boreal spring season with a mean mass concentration of 21.59 µg m$^{-3}$, and lowest was in fall with a mean concentration of 8.44 µg m$^{-3}$. Three-day wind back trajectory analysis of air mass reveals the influence of the North Western Pacific monsoon regimes on PM$_{2.5}$ concentration. During southwest monsoon, sea salt is the dominant component of fine aerosols carried by moist air from the South China Sea. During northeast monsoon, on the other hand, both wind and receptor model (USEPA PMF) analysis showed that higher particulate concentration was due to the Long Range Transport (LRT) of anthropogenic emissions from the northern East Asia. Overall, sea salt and soil comprise 33% of total PM$_{2.5}$ concentration while local biomass burning makes up 33%. LRT of industrial emission, solid waste burning and secondary sulfate from East Asia have a mean contribution of 34% to the total fine particulate for the whole sampling period.

## 1 Introduction

Increasing industrial emission and open burning of biomass and solid waste have shifted much interest in the local, regional and global transport of aerosol in Asia (Akimoto, 2003; Smith et al., 2011; Wang et al., 2014; Huang et al., 2013; Field and Shen, 2008). Aerosols are known not only for their impacts on health (Pope et al., 2002; Lelieveld et al., 2015; Jerrett, 2015; Silva et al., 2013), but also on its effects on Earth's energy budget (Ramanathan et al., 2013; Hansen et al., 1997; Chung et al., 2010). Its influence on the climate remains as one of the main uncertainties in our understanding of the atmosphere (IPCC, 2007). Rapid industrialization and urban development in the recent decades in Asia, particularly in mainland China, has led to an increase in energy consumption and consequently pollutant emissions. High emissions from the Asian main continent that is transported to other East Asian countries can contribute to elevated concentrations of ambient fine particulate concentration elsewhere (Oh et al., 2015; Gu et al., 2016; Lin et al., 2014; Zhu et al., 2017; Cayetano et al., 2011). Similarly, biomass burning from land clearing in countries like Indonesia and Malaysia has also affected their neighboring countries with reduced visibility and poor air quality (Aouizerats et al., 2015; Forsyth , 2014). Such events can have significant social, political and economic impacts on the region.



Aside from pollutant emission, meteorology plays a significant role in transboundary pollution. Certain weather patterns create transport pathways for Long Range Transport (LRT) of gases and aerosols in the atmosphere. Outflow patterns of dusts and air pollutants can be induced from frontal lifting ahead of a southwestward moving cold front (Liu et al., 2003) or from two sequential low pressure systems interacting with a tropical warm sector (Itahashi et al., 2010) in East Asia. Also, the "warm conveyor belt" mechanism causes the seasonal uplifting and eventual transport of aerosols from East Asia to the free troposphere towards the Northwest Pacific (NWP) region (Eckhardt et al., 2004). Towards the Southeast Asian (SEA) region, biomass burning in the maritime continent peaks during the spring season and is modulated by multi-scale meteorological factors like El Niňo Southern Oscillation (ENSO), Inter-Tropical Convergence Zone (ITCZ) position, Indian Ocean Dipole (IOD), Madden-Julian Oscillation (MJO), monsoon winds and Tropical Cyclones (TC) (Field and Shen, 2008; Reid et al., 2015). Its effects cover large regions of SEA. In certain instances, even reaching southern China and Taiwan (Lin et al., 2007, 2013). The life cycle of these aerosols and its impacts on the regional climate system is the subject of several field campaigns across the region (i.e. 7-Seas (7-South-East Asian Studies), CAMPEX (Clouds, Aerosol, and Monsoon Processes-Philippines Experiment), YMC (Year of the Maritime Continent ), BASE-ASIA (Biomass-burning Aerosols in South-East Asia: Smoke Impact Assessment), Fire Locating and Monitoring of Burning Emissions (FLAMBE)).

The region's complex meteorology, warm ocean water, high sensitivity to climate change and abundant aerosol sources create a complex aerosol-cloud-climate interaction that is still not well understood (Yusuf and Francisco, 2009; Reid et al., 2013). To the west of the Philippines, Reid et al. (2015) observed that the large-scale aerosol environment in the South China Sea (SCS) is modulated by MJO and TC activity in the NWP basin. TCs induce significant convective activity throughout the SCS that can extend for thousands of kilometers. The associated rainfall were seen as an effective means in aerosol scavenging that leads to low aerosol concentration despite numerous sources of emission in the region. Alternatively, high aerosol concentrations were observed along western Philippines during drier periods. To the east of the Philippines, the Pacific "warm pool" is among the warmest ocean area in the world (Comiso et al., 2015). The warm pool is also the main source of regional stratospheric air (Fueglistaler et al., 2004). A study by Rex et al. (2014) reported the existence of a pronounced minimum in columnar ozone, as well as tropospheric column of the radical OH in the warm pool region. This will have implications on the global climate system as climate change may lead to an even warmer warm pool (Comiso et al 2015), and at the same time likely modify the abundance of OH (Hossaini et al., 2012). These factors may contribute in prolonging the lifetime of biomass burning-induced pollutants that can increase stratospheric intrusion in the future. Moreover, monsoon wind flows that influence regional climate and weather patterns also modulate aerosol transport. Using a chemical transport model WRF-CHEM, Bagtasa (2011) found that the two main monsoon regimes, northeast and southwest monsoon, mostly isolate the Philippines from East Asian pollution. However, northwesterly winds that can transport pollutants from southern China can be induced by TCs during its passage to the north or northeast of the Philippines.

There is limited literature on LRT aerosol observation in the country. This is perhaps due to the geographic separation of the Philippine archipelago from the Asian continent. Nevertheless, the Philippines have been identified as a source of biomass burning emissions ubiquitous in SEA. Gadde et al. (2009) estimated an annual open field burning of 10.15 Tg of rice straw from 2002-2006 in the Philippines. Leading to observed elevated levels of levoglucosan and Organic Carbon (OC) at several





sampling sites in Hong Kong during the springtime of 2004 and summer of 2006 where air parcels originating from the Pacific passed through the northern island of the Philippines (Sang et al., 2011; Ho et al., 2014). The lack of observations in the northern Philippine region between the East Asian subtropics and the maritime continent of SEA makes this location a blind-spot in our knowledge of the current state of atmospheric environment. Also, satellite-based observations are hindered by
persistent cloud cover over the region.

This study aims to characterize the chemical composition of PM$_{2.5}$ on the northwest region of the Philippines, identify source contributions using a receptor model and investigate existing transport pathways in the NWP region. This paper will be presented as follows: the next section will detail the characteristics of the sampling site, aerosol sampling methodology, wind back trajectory and receptor modelling. The third section will discuss the influence of the NWP/Asian monsoon on the
seasonal variability of observed concentrations of fine aerosol mass and its components, as well as emission sources derived from meteorological and chemical receptor modeling. Finally, the last section will summarize the results of this study.

## 2   Methodology

### 2.1   Sampling site

Burgos (18.5° N, 120.57° E), a small town in the province of Ilocos Norte, is located in northwestern Luzon, northern Philip-
pines as shown in figure 1. A filter-based air sampler (BGI PQ200, USA) was placed approximately 12 m above ground level atop a 3-storey building. The site is a rural environment surrounded by vegetation where the SCS (locally known as the West Philippine Sea) is 500 m to the west and a range of hills approximately 700 m to the east. A nearby road 100 m to the east is present, but has low daily traffic volume.

Burgos is classified as a Type 1 climate under the modified Coronas climate type classification (Coronas, 1912) where the
region experiences wet season from May to September and a distinct dry season from October to April. Sampling during summer (August - September) of 2015 coincided with a monsoon break, thus all sampling days for all seasons in this study were non-rainy days. The area is also characterized by high winds during the boreal winter season that is mainly attributed to the cornering effect of the northeast monsoon winds to Luzon island.

### 2.2   Sample collection

Daily PM$_{2.5}$ samples were collected in August to September 2015, November 2015, January to February 2016 and March 2016 to represent the boreal summer, fall, winter and spring, respectively. Two-week (14 days) sampling was conducted for each season. Except for the summer sampling period when the northern region of the Philippines suffered provincial-wide power failure due to the effects of Typhoon Goni (locally named "Ineng"). Only seven days of sampling was done for summer. Table 1 summarizes the sampling dates of this study. Samples were collected using a 47 mm quartz fiber filter at a flow rate of 16.7
Lmin$^{-1}$ from 1000H Philippines Standard Time (PST; +8 UTC) to 1000H PST the following day.



### 2.3 Chemical analysis

Prior to sampling, the quartz fiber filters are pre-heated at $900°$ C for 1.5 hours to remove impurities. Each filter is then weighed before and after sampling using a microbalance (Satorius MC5). The filter is then cut into four identical parts: One for the analysis of carbonaceous components, other parts for water-soluble ionic species, for metallic elements and for the analysis

of anhydrosugar.

Carbonaceous contents of $PM_{2.5}$ were measured using an elemental analyzer (Carlo Erba, Model 1108). The quarter part filter was divided into two, one part was heated with hot nitrogen gas ($340$-$345°$ C) for 30 minutes to remove the OC fraction while the other part was analyzed without heating, the filter was then fed to the elemental analyzer to determine the amount of elemental carbon (EC) and total carbon (TC), respectively. OC concentration was calculated by getting the difference of

TC and EC. Another filter quarter was placed in a 15 ml polyethylene (PE) bottle filled with distilled and deionized water and subjected to ultrasonic extraction for 60 min, maintained at room temperature. Ion chromatography (DIONEX DX-120) was utilized to analyze the major anions ($F^-$, $Br^-$, $Cl^-$, $SO_4^{2-}$ and $NO_3^-$) and cations ($NH_4^+$, $Ca^{2+}$, $Na^+$, $K^+$ and $Mg^{2+}$).

The last part of the filter was digested with a 30 ml mixed acid solution ($HNO_3$:$HCLO_4$, 3:7) at $150$-$200°$ C. After which the solution was diluted with 25 ml distilled and deionized water and stored in a PE bottle. Metallic elements (Al, As, Ca,

Cd, Cr, Cu, Fe, K, Mg, Mn, Ni, Pb, Ti, V and Zn) were determined using an Inductively Coupled Plasma-Atomic Emission Spectrometer (ICP-AES, Perkin Elmer, Optima 2000DV).

### 2.4 Wind and receptor modeling

Analysis of wind back trajectories was done using the HYSPLIT model (Draxler and Hess, 1998). Meteorological conditions were driven by output from Weather Research and Forecast (WRF) model (Skamarock et al., 2008) run with 5-day spin up

time for each sampling period. WRF model with spectral nudging was used to downscale the FNL final reanalysis (downloaded from http://rda.ucar.edu) from $1°$ X $1°$ horizontal spatial resolution to 15 km resolution in a two-way nested domain of 45-15 km grid resolution. Twice-daily 72H back trajectories were then plotted and grouped into five clusters representing the general area of wind sources.

Receptor models are used to quantify the levels of air pollution, disaggregated into sources using statistical analysis of

particulate matter concentrations and its chemical components. Positive Matrix Factorization (PMF) is a widely used receptor model by the US Environmental Protection Agency (US EPA). The US EPA PMF has been applied to identify and apportion the air pollution sources in an industrial district of the capital city of Metro Manila, Philippines, in which lead (Pb) was found to have significant contributions in both the coarse ($PM_{2.5-10}$) and the fine ($PM_{2.5}$) particulate matter fractions (Pabroa et al., 2011). PMF is utilized in this study to identify possible emission sources of observed fine aerosols.





## 3 Results and Discussion

### 3.1 Monsoon winds

The Philippines is categorized as a tropical rainforest / monsoon climate in the Köppen-Geiger climate classification. Its seasons are mainly described as wet or dry. The seasonality used in this study mainly refers to the prevailing winds of the NWP/Asian

monsoon, rather than changes in local temperature and rainfall as used in other climate classification methods. Figure 2 a-d show the prevailing winds (arrow), accumulated rainfall (shading) and twice-daily 72H wind back trajectory (red line) in the NWP region during each of the four sampling periods. Averaged wind vectors are from the 6-hourly NCEP FNL reanalysis data and accumulated rainfall is from the TRMM 3B42A version 7 rainfall data product. Back trajectories are derived from the HYSPLIT-WRF simulation.

In the months of June-July-August (hereafter written as JJA, same for other seasons) or the boreal summer season, southwest monsoon wind prevails over western Philippines as shown in fig. 2a. The southwest monsoon period usually starts in the latter part of May and ends in September (Moron et al., 2009). The monsoon wind brings in warm moist air from the SCS making the western coasts of the Philippines wet this season (Flores and Balagot, 1969; Bagtasa, 2017). SON or fall season in fig. 2b is marked by the southeast propagation of the ITCZ which results in the shifting of monsoonal winds from southwest to easterlies

in September, then a northeasterly direction by the end of October (Bagtasa, 2017). Figure 2c shows northeast winds prevail during DJF or boreal winter. This season is also characterized by rainfall along the eastern coastal regions of the Philippines (Akasaka et al., 2007). And in fig. 2d, MAM (spring) marks the transition between northeast and southwest monsoon regimes. In this period, most convection stays near and south of the equator (Chang et al., 2005) with prevailing northeast to easterly winds from the Pacific Ocean.

Climatologically, the northeast to southwest monsoon transition starts in the middle of March, but a late winter monsoon surge coincided with the spring sampling period. Moreover, the years 2015 to early 2016 are strong El Niňo years, however, its influence on the NWP monsoon system will not be further discussed.

### 3.2 Seasonal variation of PM$_{2.5}$

The 24H PM$_{2.5}$ mass concentration is shown fig. 3. It has a strong seasonal variation, where the lowest mass concentration

(5.7 $\mu$g m$^{-3}$) is seen in fall and the highest (32.3 $\mu$g m$^{-3}$) in spring time. PM$_{2.5}$ mass concentration for summer, fall, winter and spring had an average value and standard deviation of 11.9 ± 5.0 $\mu$g m$^{-3}$, 8.4 ± 2.3 $\mu$g m$^{-3}$, 12.9 ± 4.6 $\mu$g m$^{-3}$ and 21.6 ± 6.6 $\mu$g m$^{-3}$, respectively. The results show comparable concentrations measured from Dongsha island in northern SCS except for heavy aerosol events previously reported in that site (Atwood, 2013; Lin et al., 2013). However, we expect its sources to vary from our observations based on the MODIS-derived AOT analysis of Lin et al. (2007) which showed the northern SCS

to be significant influence by southern and eastern China emissions. Carbonaceous components EC and OC in fig. 4 generally followed the same seasonal variation. Minimum concentration was observed in fall (EC 0.40 ± 0.09; OC 0.63 ± 0.18) in $\mu$g m$^{-3}$ and maximum in spring (EC 1.03 ± 0.21; OC 1.76 ± 0.39) in $\mu$g m$^{-3}$. The annual EC and OC mean concentration and standard deviation is 0.67 ± 0.3 $\mu$g m$^{-3}$ and 1.15 ± 0.63 $\mu$g m$^{-3}$, respectively. Measured EC likely originated from diesel





buses and trucks that pass by the adjacent road. Traffic volume does not vary much near the sampling site which explains the small standard deviation observed. Overall, total carbon contribution to $PM_{2.5}$ is 13.4% ±3.5%.

The mass ratio of the carbonaceous components OC/EC has been previously shown to determine contribution from primary or secondary sources (Chow et al., 1996). Figure 5 shows the seasonal average OC/EC mass ratio is 1.42, 1.74, 1.71 and 1.79

for summer, fall, winter and spring season, respectively. The bold dashed line represents ratio value of 2. Mean OC/EC ratio for all seasons are below 2 which indicates that fine particulates are dominated by primary aerosol (Chow et al., 1996). On the basis of individual days, however, a third of the winter and spring data had values of OC/EC greater than 2.

### 3.2.1 Southwest Monsoon (Summer)

Upwind regions during southwest monsoon are known large aerosols emitters, particularly from biomass burning. However,

there is low observed aerosol concentration in this season. This is likely due to active convection around the island nations across the SCS (i.e. Borneo, Indochina peninsula) during the sampling period. Most air parcels indicated by the wind back trajectories originated from the marine boundary layer near these island regions. These air parcels were then transported along the western coast of northern Luzon before reaching the sampling site. The western coast of Luzon is characterized by substantial precipitation during the southwest monsoon season as a result of moist air being orographically lifted by the

Cordillera mountain range in western Luzon (Cayanan et al., 2011). An average accumulated rainfall of 91.2 mm was recorded along northwestern Luzon coast throughout the seven day summer sampling period. These factors would have resulted in the suppression of biomass burning in the SEA region and scavenging of particulates along the path of the transported air parcels. In addition, the WRF simulation used to drive the HYSPLIT model showed a strong diurnal cycle of land/sea breeze along the western Luzon coasts. Nighttime land breeze carries polluted air from central and southwestern Luzon northwestward

towards the SCS through Lingayen gulf. After which, daytime sea breeze pushes back these polluted air masses inland along the northwest Luzon region.

### 3.2.2 Southwest to Northeast Monsoon transition (Fall)

November (fall) sampling showed the lowest mass concentrations among all seasons. During the fall monsoonal transition regime, easterlies bring in air mass from the northwest Pacific Ocean (shown in fig. 2b) where no known large emission

sources are present. Contribution from the eastern region of northern Philippines appears to be minimal as the northwest and northeast Philippines are separated by the northern hills of the Cordillera mountain range. Also, northeast Philippines is mainly composed of agricultural land with only little to moderate urban activity.

### 3.2.3 Northeast Monsoon (Winter and Spring)

Highest mass concentration is seen during spring time, followed by the winter observation. Strong northeasterly wind affected

both sampling periods. Wind back trajectories of both seasons in fig. 2c and 2d show air parcels come from northern East Asia. However, better outflow patterns of pollutants from northern Asia during springtime may have contributed to higher



observed mass concentrations in March. In addition, heavier precipitation from the Meiyu/Baiu front located along the East Asian subtropics during winter, as seen in fig. 2c, likely reduced the transported aerosols by wet scavenging before reaching the Philippines. The chemical characteristics and possible aerosols sources will be discussed in the succeeding sections.

### 3.3  Ionic and metallic components

Seasonal mean and standard deviation of $PM_{2.5}$ and some water soluble ionic components are shown in fig. 6. Figures 6b, 6c and 6d show $NO_3^-$, $SO_4^{2-}$ and $NH_4^+$, respectively, also follow the seasonal variation of $PM_{2.5}$ mass concentration. Minimum concentration was observed in fall and maximum during spring sampling. These three components are associated with secondary inorganic aerosol and make up on average $69 \pm 4\%$ of the total water soluble ions. Among all ionic species, $SO_4^{2-}$ has the highest contribution at $2.4 \pm 0.4\,\mu g\,m^{-3}$, followed by $NO_3^-$ at $1.0 \pm 0.3\,\mu g\,m^{-3}$ and $NH_4^+$ at $0.7 \pm 0.1\,\mu g\,m^{-3}$. Seasonality of $Na^+$ with mean concentration of $0.4 \pm 0.1\,\mu g\,m^{-3}$ and $Ca^{2+}$ ($0.3 \pm 0.1\,\mu g\,m^{-3}$) also shows the same seasonal variability. However, this seasonality is not apparent for the ions $Cl^-$, $K^+$ and $Mg^{2+}$.

In fig. 6f, chlorine ($Cl^-$) summer sampling show highest concentration of $0.69 \pm 0.1\,\mu g\,m^{-3}$ and the rest of the seasons with nearly constant concentration of $0.52 \pm 0.09\,\mu g\,m^{-3}$ for fall, $0.57 \pm 0.07\,\mu g\,m^{-3}$ for winter and $0.60 \pm 0.12\,\mu g\,m^{-3}$ for spring. We attribute the high $Cl^-$ content in summer to sea salt carried by the southwest monsoon wind. Potassium ($K^+$) with mean concentration of $0.23 \pm 0.06\,\mu g\,m^{-3}$ is a relatively abundant element in crustal rocks (Mason, 1966) and is also used as tracer for wood burning due to the significant amount of $K^+$ in wood biomass (Miles et al., 1996). Compared with levoglucosan measurements shown table 2, $K^+$ is highly correlated to levoglucosan with correlation coefficient $r = 0.99$ at 95% confidence interval ($p < 0.05$). This indicates that measured $K^+$ is mainly from open burning of biomass, which is more widespread during the dry season of winter and spring. Magnesium ($Mg^{2+}$) has maximum concentration in spring and lowest in winter. For summer and fall, $Mg^{2+}$ is highly correlated with $Ca^{2+}$ (summer $r = 0.94$; fall $r = 0.97$, both at $p < 0.05$), on the other hand, no significant correlation were found in other seasons. This suggests that the source of $Mg^{2+}$ is mostly from mineral dust (carbonate mineral) when highly correlated with $Ca^{2+}$ (Li et al., 2007). The high concentration of $Mg^{2+}$ in spring is therefore attributed to non-local sources. Source attribution for these ions are further discussed in the results of receptor modeling in the proceeding section.

Figure 7 shows some metallic components of measured fine particulates. Metallic components Al and Fe in figs. 7a and 7d, respectively, are associated with crustal origins (Mason, 1966), The seasonal concentrations of which are summarized in Table 2. Seasonal variation of heavy metals Cd, Cr, Ni and Pb are shown in fig. 7b, 7c, 7e and 7f, respectively. Heavy metal components of fine particulates pose a health risk (Monaci et al 2000). Particularly, Ni, Cd and Cr are identified as human carcinogens while Pb is toxic and exposure can lead to permanent adverse health effects in humans (WHO, 1994). Dispersion of metals embedded in particulates also determines the rate at which metals deposit on Earth's surface (Allen et al. , 2001). All heavy metal components were evidently high in spring, followed by the winter sampling period. Ambient concentration of anthropogenic components depend on distance from source location and transport process (Thomaidis et al., 2003). Since no large industries or power plants within $250\,km$ of the sampling site are present, these toxic components likely originated from upwind regions during northeast monsoon. No significant correlations were found between these metallic components with



ionic components associated with secondary inorganic aerosols. This suggests that these heavy metal components come from several different sources. Table 2 is the summary of the seasonal mean mass concentration and their corresponding standard deviation of $PM_{2.5}$ and its components, including the anhydrosugar levoglucosan.

Figure 8a shows the ratio of cation and anion close to unity for all seasons (Figure 8 is in units of equivalent concentrations). This indicates good charge balance of atmospheric aerosols and high data quality used in this study. Figure 8b shows the scatter plot of $Cl^-$ versus $Na^+$. The ratio $Cl^-/Na^+$ shows highest value in summer with 1.19, 1.08 for fall, 0.99 for winter and 0.89 for spring. The ratio indicates that summer $Cl^-$ is mainly from sea salt where mean $Cl^-/Na^+$ ratio of sea salt is equivalent to 1.17 (Chester et al., 1990). This result supports our initial hypothesis that the high summer $Cl^-$ concentration mainly comes from sea salt. In addition, there may be $Cl^-$ depletion in the rest of the seasons due to the following factors: 1) farther distance from upwind coast (Dasgupta et al., 2007), 2) high sulfate and nitrate concentration during northeast monsoon may have reacted with $Cl^-$ in sea salt forming gas phase HCl in the process (Virkkula et al., 2006), and 3) excess $Na^+$ may have come from resuspended soil due to stronger wind in non-summer seasons. This is further supported by the high correlation values found between Fe and Al in winter (r = 0.71 at p < 0.05) and spring (r = 0.88 at p < 0.05) measurements which suggests higher loading of uplifted dust blown by strong winds during those seasons.

Also mentioned in the previous section, $Mg^{2+}$ is highly correlated with $Ca^{2+}$ for summer and fall. This indicates mineral dust as main source of $Mg^{2+}$ in these seasons. In terms of the ratio of $Mg^{2+}/Na^+$, among all seasons, winter shows closest to the mean sea salt ratio of 0.23 (Chester, 1990), indicating mostly non-sea salt source for Mg except winter. Furthermore, ratio of both $Mg^{2+}/Ca^{2+}$ and $Mg^{2+}/Na^+$ tends to vary more and spread out during spring season sampling as seen in graphs of fig. 8c and 8d. For the components associated with secondary inorganic aerosols, fig. 8e shows the ratio of $NH_4^+/SO_4^{2-}$ all below unity (bold dashed line). The ratio points to $NH_4^+$ not fully neutralizing $SO_4^{2-}$ all throughout the sampling periods. This is possibly due to nearby sources of sea salt sulfate (S.S. $SO_4^{2-}$) in the region. Similarly, the ratio $NH_4^+/[NO_3^- + SO_4^{2-}]$ are also found to be below unity for all seasons. This suggests the presence of $NH_4^+ NO_3^-$ as well as other forms of $NO_3^-$ in the region.

## 3.4 Source contribution

The US EPA PMF 5.0 was used to resolve the contribution of the identified factor sources to the $PM_{2.5}$ concentration on each sampling day. The US EPA 5.0 uses a weighted least squares model, weighted based on known uncertainty or error of the elements of the data matrix (Paatero, 1999). The goal is to obtain the minimum Q value after several iterations, keeping the residuals at the most reasonable levels and having a sensible and rational factor profile. Details of the US EPA 5.0 are described elsewhere (Paatero, 1999; Paatero et al., 2002). In this study, all 49 sampling datasets were used to resolve the factor and contribution profiles of $PM_{2.5}$ in northwestern Philippines. An extra 10% modeling uncertainty was added to the data to obtain the desired, optimum convergence of the Q, and acceptable scaled residuals in the run.

Here, we obtained 6 source factors namely: 1) sea salt, 2) resuspended fine dust, 3) local solid waste burning, 4) LRT of industrial emissions, 5) LRT solid waste burning and 6) LRT secondary sulfate. Figure 9 shows the profiles of the factor (sources) identified. Figure 10 shows the daily contribution per season for each of the source profiles. Using the source contributions, we were able to resolve the seasonal concentration of the sources, consistent with the factor profiles and fingerprints. For instance,





elevated levels of sea salt contributed mainly during the summer season (5.7 ± 1.5 µg m$^{-3}$), consistent with our analysis. On the other hand, LRT industrial emissions are observed at elevated levels during spring (4.0 ± 3.8 µg m$^{-3}$) and winter seasons (3.1 ± 2.7 µg m$^{-3}$), consistent with the wind back trajectory analysis discussed. Table 3 summarizes the seasonal and total contribution of source factors to fine particulate matter of the region. Overall, natural primary sources sea salt and resuspended

fine dust constitutes 33% of atmospheric aerosols. Another 33% or one third is due to local solid waste burning. This includes open burning of biomass in the dry season for the purpose of land clearing ubiquitous in SEA. Lastly, 34% is due to LRT sources from industrial emission, solid waste burning and secondary sulfate.

### 3.5   Enrichment factor

The enrichment factors of the chemical markers of the identified sources are tabulated in Table 4. Factors associated with solid

waste burning are divided into local and LRT burning factors. Both have high associations with K$^+$, Zn and OC. The LRT solid waste burning factor exhibits strong association with NO$_3^-$, NH$_4^+$, Mg$^{2+}$, Ca$^{2+}$, K$^+$, Zn, OC and EC. The enrichment factors of OC, EC, Zn and NO$_3^-$ with respect to K$^+$ for local burning decreased to 50% when compared to the LRT counterpart, indicating the decrease in ageing of the PM$_{2.5}$ components as particles are transported over a long distance. The two other LRT factors identified, secondary sulfate and industrial factor source, showed strong associations with the heavy metals Cr, Ni, Cu,

Cd and Pb. These chemical markers are reported in petroleum, chemical and manufacturing industries (Park et al., 2002) that are not locally present. The secondary sulfate source marked an enrichment factor for (Ca$^{2+}$+Mg$^{2+}$)/Na$^+$ of 5.7, which is about the value of enrichment factor of a certified reference material of China loess soil (Nishikawa et al., 2000), while that of the industrial emission factor (1.6) corresponds to the enrichment factor of sea salt ageing on the processed dust particles from a marine background site in Korea (Cayetano et al., 2011).

It is noteworthy that significant contribution from long distance sources are observed during the northeast monsoon seasons of winter and spring. Analysis of wind back trajectory, PMF model and chemical components all demonstrate the existence of transboundary aerosols by way of the northeast monsoon wind. However, relatively lower concentration of components linked to LRT found in winter is likely modulated by rainfall associated with the Meiyu/Baiu front (shown in fig. 2c). More (less) frontal rain in winter (spring) resulted in increased (decreased) aerosol scavenging, which affected the overall transport

flow of LRT fine aerosols. Figure 11 shows high correlation (r = 0.87) between the observed and reconstructed PMF-modeled PM$_{2.5}$mass concentration. Providing high confidence on the PMF analysis.

### 4   Conclusions

This study describes the seasonal characteristics of fine particulates (PM$_{2.5}$) in Burgos, Ilocos Norte, located in the northwestern edge of the Philippines. This region is located between the East Asian subtropics and the maritime continent of SEA. Both

regions are known emitters of large quantities of anthropogenic aerosols. Observed fine particulates are mainly modulated by the NWP monsoon winds. PM$_{2.5}$ shows strong seasonality where the lowest mean concentration is found during fall season when easterly winds prevail. High concentrations were found in winter and spring time during the northeast monsoon season.

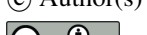



Components of fine particulates also showed distinct seasonality. Carbonaceous aerosol components EC and OC have an annual mean value of $0.67 \pm 0.3\ \mu g\,m^{-3}$ and $1.15 \pm 0.63\ \mu g\,m^{-3}$, respectively. Both lowest in fall and highest in spring. EC and OC collectively make up $13.4 \pm 3.5\%$ of observed fine particulates. The ionic and metallic components of fine aerosols also varied by sampling period which generally followed the seasonal variation of $PM_{2.5}$, and make up $44.4 \pm 10.1\%$ and
$11.7 \pm 3.8\%$ of $PM_{2.5}$ mass, respectively. Analysis of the chemical components reveals high sea salt content in summer when southwest monsoon winds prevail, and high concentration of components associated with secondary inorganic aerosols (i.e. $NO_3^-$, $SO_4^{2-}$ and $NH_4^+$) as well as anthropogenic pollutants (i.e. heavy metals) during northeast monsoon. HYSPLIT-WRF wind back trajectory results show air masses originating from East Asia moves along the northeasterly wind in winter and spring seasons. Winter sampling showed comparatively lower concentrations of $PM_{2.5}$ than spring. We attribute this to the
scavenging of transported aerosols by the Meiyu/Baiu front, which had higher precipitation during the winter sampling period.

Positive Matrix Factorization (PMF) of the US EPA was used to determine the source contributors of fine particulate in the region. The results of the PMF receptor model and wind analysis were consistent and complementary. Here, six source profiles were obtained using the receptor model, namely: 1) sea salt, 2) resuspended fine dust, 2) local solid waste burning, 4) LRT of industrial emissions, 5) LRT solid waste burning and 6) LRT secondary sulfate. Consistent with the chemical analysis, high
sea salt in summer contributes to almost half of aerosol content for that season. Resuspended fine dust is seen to increase in the spring and winter season when strong winds prevail over the sampling region. Open burning of biomass and solid waste is widespread in the dry seasons of winter and spring. This is seen in the seasonality of $K^+$ and the anhydrosugar levoglucosan, which were found to be highly correlated with one another. LRT of anthropogenic fine particulates were observed during winter and spring time when the northeast monsoon serves as transport pathway for East Asian aerosols to reach the northern part of
the Philippines. The annual mean source contribution of transboundary industrial emission, secondary sulfate and solid waste burning was 14%, 9% and 11%, respectively. In total, LRT contributes to one third of aerosol content in the region.

Figure 12 shows all the wind back trajectories grouped into five clusters. 17% (pink) of all simulated wind back trajectories occurred during southwest monsoon shows air parcels originated from the SCS and moved along the western coast of northern Luzon. In fall, 27% (blue) and 14% (red) originated from the near and far east Pacific, respectively. Total of 41% of pristine
air parcels originated from the Pacific waters. This period is when lowest $PM_{2.5}$ mass concentration including most of its components was observed. During northeast monsoon, 30% comes from northern East Asia and another 13% from east China sea. All five back trajectory clusters were simulated from ground to ground transport.

To our knowledge, this is the first comprehensive analysis of aerosol characteristics in this region of the Philippines. Also, this is the first study to confirm long range transport of East Asian aerosols to the country. It would be interesting to see its
implications on the region's radiative forcing, aerosol-cloud-climate interaction and stratospheric intrusion, if there are any. These are questions essential to a better understanding of the region's atmosphere.

*Competing interests.* No competing interests are present



*Acknowledgements.* The authors would like to acknowledge the Department of Science and technology (Philippines) and the Ministry of Science and Technology (Taiwan) for funding the project entitled "Tempospatial Distribution and Transboundary Transport of Atmospheric Fine Particles Across Bashi Channel, Taiwan Strait and South China Sea".





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





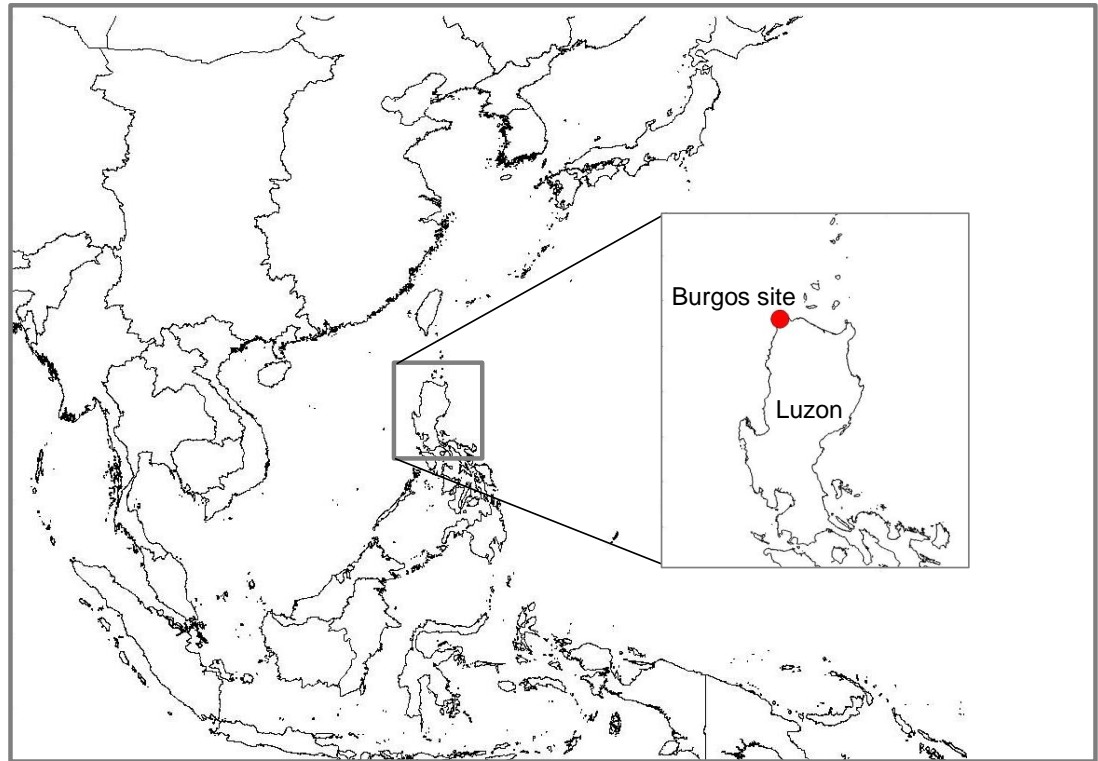

**Figure 1.** Map of East Asia, Southeast Asia and sampling site in Burgos, Philippines

**Table 1.** Sampling dates

| Season | Sampling dates | Sampling days |
|--------|----------------|---------------|
| Summer | Aug. 27 - Sept. 2, 2015 | 7 |
| Fall | Nov. 5 - Nov. 18, 2015 | 14 |
| Winter | Jan. 21 - Feb. 3, 2016 | 14 |
| Spring | Mar. 17 - Mar. 30, 2016 | 14 |

Zhu, J., Liao, H., Mao, Y., Yang, Y., and Jiang, H.: Interannual variation, decadal trend, and future change in ozone outflow from East Asia,
    Atmos. Chem. Phys., 17, 3729–3747, doi:10.5194/acp-17-3729-2017, 2017.







**Figure 2.** Wind (arrows), accumulated rainfall (shading, in mm) and wind back trajectory (red line) during a) summer, b) fall, c) winter and d) spring sampling period.





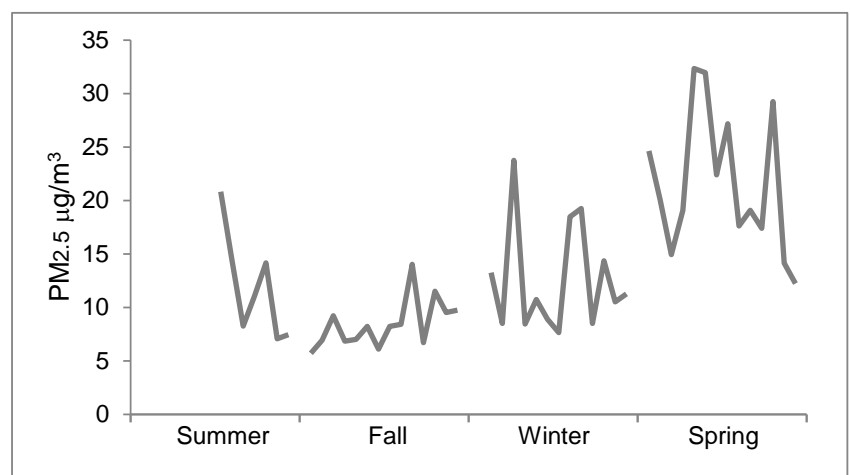

**Figure 3.** Daily and seasonal variation of PM$_{2.5}$ mass concentration.

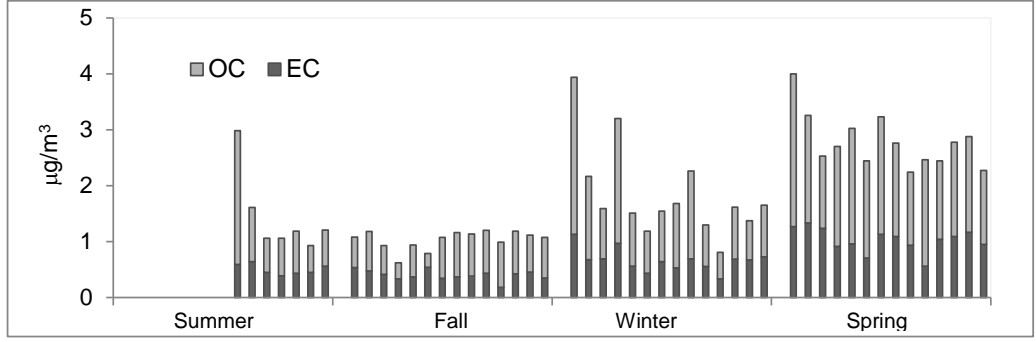

**Figure 4.** Daily Elemental Carbon (EC) and Organic Carbon (OC) mass concentration.





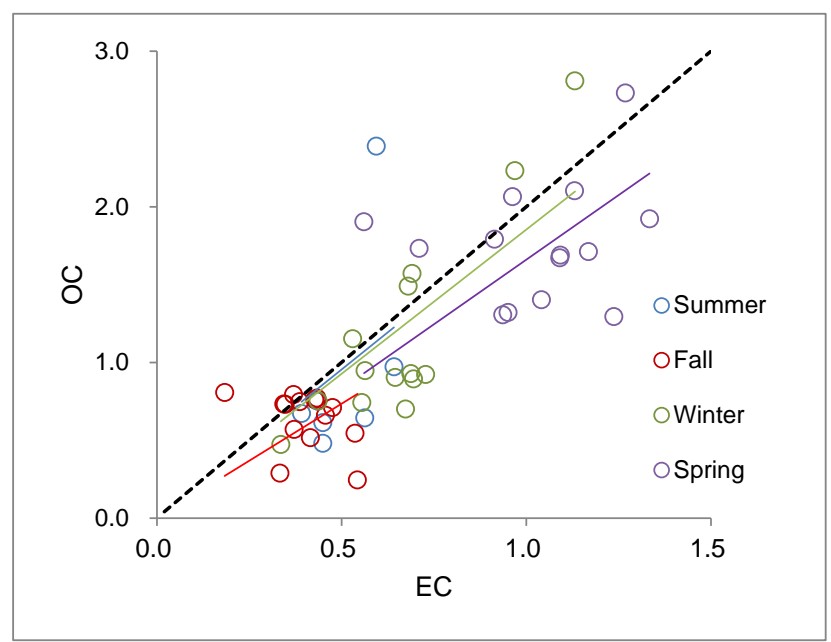

**Figure 5.** Scatterplot of OC and EC for summer, fall, winter and spring sampling period (in $\mu g\, m^{-3}$).





**Figure 6.** Fine particulate ionic components mass concentration (unit in $\mu g\,m^{-3}$).





**Figure 7.** Fine particulate metallic components mass concentration (unit in $\mu g\, m^{-3}$).





**Figure 8.** Scatter plot of a) total anion and cation, b) $Na^+$ and $Cl^-$, c) $Mg^{2+}$ and $Na^+$, d) $Mg^{2+}$ and $Ca^{2+}$, e) $NH_4^+$ and $SO_4^{2-}$, and f) $NH_4^+$ and $NO_3^- + SO_4^{2-}$ (unit in equivalent concentration ).





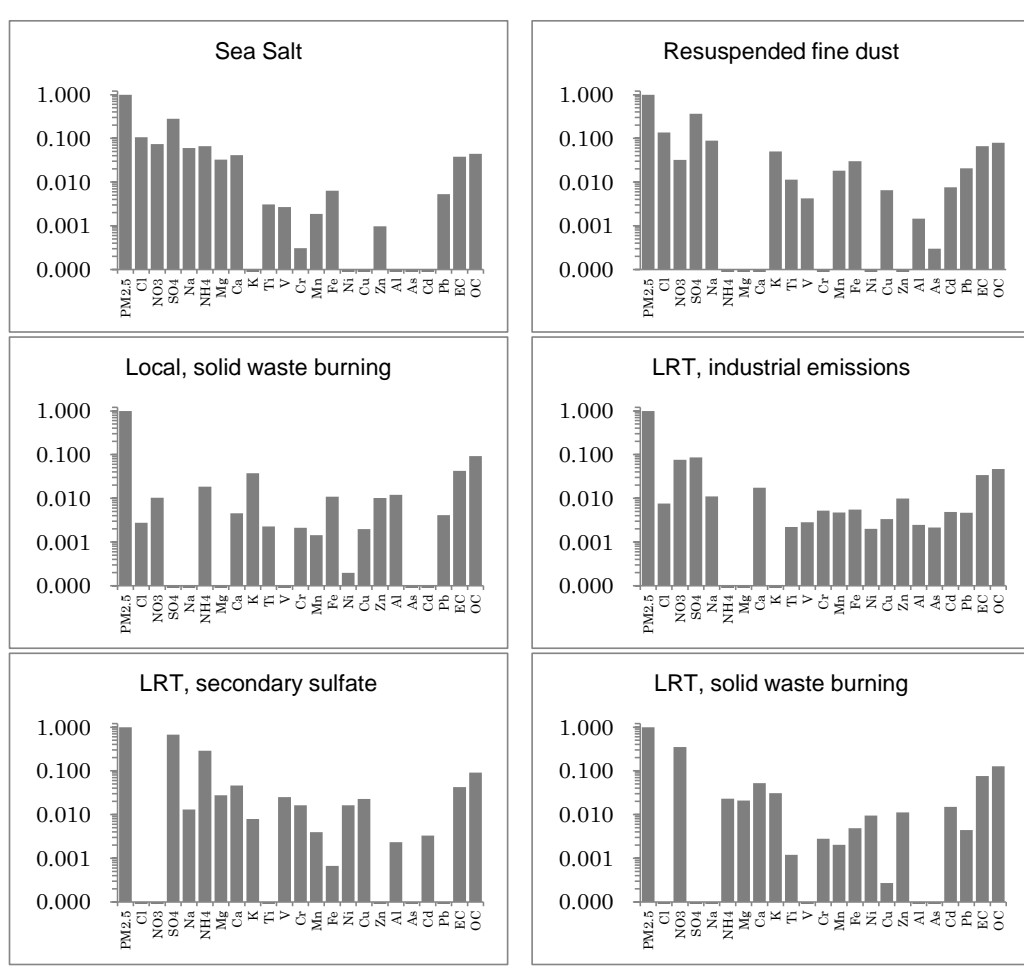

**Figure 9.** Source factor profiles from PMF analysis.





**Figure 10.** Daily contribution for each source profile for all sampling period (unit is $\mu g\,m^{-3}$).




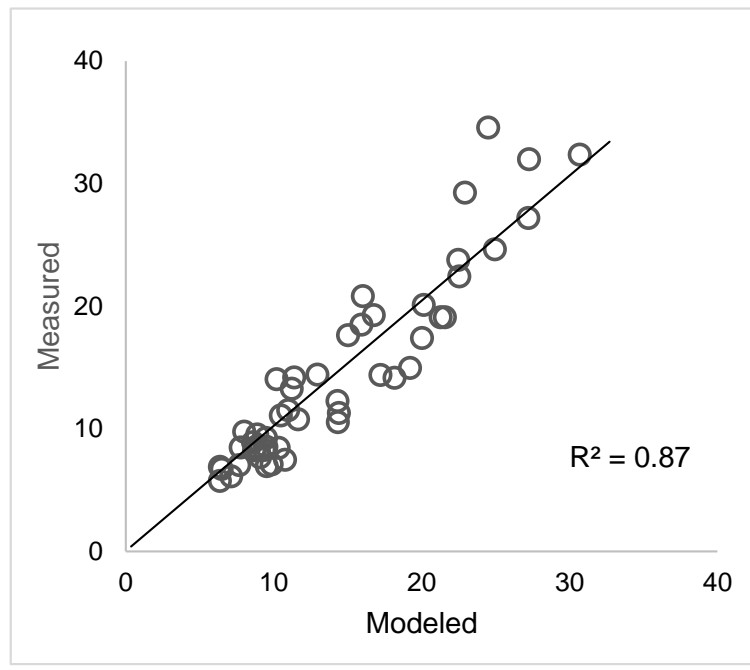

**Figure 11.** Measured vs. PMF model reconstruction (unit is $\mu g\,m^{-3}$).



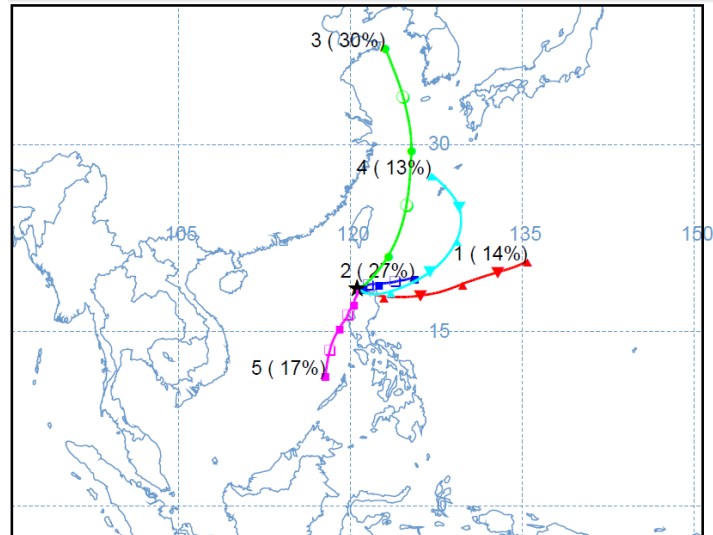

**Figure 12.** Clustered HYSPLIT-WRF wind back trajectories.



**Table 2.** Summary of mass concentration of $PM_{2.5}$ and its components (unit is $\mu g\,m^{-3}$).

|  | Summer | Fall | Winter | Spring |
|---|---|---|---|---|
| $PM_{2.5}$ | 11.9 ± 4.6 | 8.4 ± 2.2 | 14.2 ± 7.3 | 21.6 ± 6.6 |
| EC | 0.50 ± 0.09 | 0.40 ± 0.09 | 0.67 ± 0.19 | 1.03 ± 0.20 |
| OC | 0.93 ± 0.61 | 0.63 ± 0.17 | 1.18 ± 0.62 | 1.76 ± 0.37 |
| $NO_3^-$ | 0.71 ± 0.21 | 0.60 ± 0.16 | 0.80 ± 0.27 | 1.81 ± 1.29 |
| $SO_4^{2-}$ | 1.69 ± 0.43 | 1.50 ± 0.42 | 2.48 ± 0.94 | 4.06 ± 1.67 |
| $NH_4^+$ | 0.53 ± 0.15 | 0.32 ± 0.09 | 0.63 ± 0.21 | 1.35 ± 0.72 |
| $Na^+$ | 0.37 ± 0.11 | 0.31 ± 0.08 | 0.37 ± 0.15 | 0.44 ± 0.18 |
| $Cl^-$ | 0.69 ± 0.17 | 0.52 ± 0.17 | 0.57 ± 0.23 | 0.60 ± 0.24 |
| $K^+$ | 0.07 ± 0.03 | 0.08 ± 0.02 | 0.24 ± 0.07 | 0.53 ± 0.38 |
| $Ca^{2+}$ | 0.25 ± 0.05 | 0.21 ± 0.05 | 0.25 ± 0.10 | 0.63 ± 0.26 |
| $Mg^{2+}$ | 0.20 ± 0.04 | 0.14 ± 0.03 | 0.10 ± 0.02 | 0.29 ± 0.07 |
| Al | 0.05 ± 0.00 | 0.05 ± 0.01 | 0.07 ± 0.02 | 0.12 ± 0.04 |
| Fe | 0.12 ± 0.00 | 0.10 ± 0.02 | 0.15 ± 0.02 | 0.17 ± 0.02 |
| Cd | 0.01± 0.00 | 0.03 ± 0.00 | 0.04 ± 0.01 | 0.13 ± 0.04 |
| Cr | 0.02 ± 0.00 | 0.01 ± 0.00 | 0.05 ± 0.01 | 0.10 ± 0.01 |
| Ni | 0.00 ± 0.00 | 0.01 ± 0.00 | 0.03 ± 0.01 | 0.09 ± 0.02 |
| Pb | 0.06 ± 0.01 | 0.06 ± 0.01 | 0.09 ± 0.02 | 0.11± 0.01 |
| Levoglucosan* | 0.63 ± 0.86 | 1.4 ± 0.56 | 6.0 ± 2.4 | 19.1 ± 2.8 |

* unit in $ng\,m^{-3}$

**Table 3.** Summary of seasonal and annual source profile contribution (unit is $\mu g\,m^{-3}$).

| Source | Summer | Fall | Winter | Spring | Annual |
|---|---|---|---|---|---|
| Sea salt | 5.7 ± 1.5 (49%) | 3.4 ± 1.4 (40%) | 1.9 ± 1.1 (14%) | 3.3 ± 2.6 (15%) | 3.3 ± 2.1 (23%) |
| Resuspended fine dust | 0.4 ± 0.4 (3%) | 1.2 ± 0.4 (11%) | 2.1 ± 0.7 (15%) | 1.5 ± 1.2 (7%) | 1.4 ± 1.0 (10%) |
| Local solid waste burning | 5.0 ± 1.3 (43%) | 2.8 ± 1.1 (33%) | 4.8 ± 3.9 (35%) | 6.2 ± 5.1 (29%) | 4.7 ± 3.7 (33%) |
| LRT, industrial emission | 0.02 ± 0.03 (0.1%) | 0.0 ± 0.0 (0%) | 3.1 ± 2.7 (22%) | 4.0 ± 3.8 (18%) | 2.0 ± 3.0 (14%) |
| LRT, secondary sulfate | 0.1 ± 0.1 (1%) | 0.1 ± 0.1 (1%) | 1.2 ± 0.4 (8%) | 3.3 ± 2.0 (15%) | 1.3 ± 1.7 (9%) |
| LRT, solid waste burning | 0.4 ± 0.3 (3%) | 1.0 ± 0.3 (11%) | 0.7 ± 0.5 (5%) | 3.5 ± 2.8 (16%) | 1.5 ± 2.0 (11%) |





**Table 4.** Enrichment factors

| | Solid waste burning | | |
|---|---|---|---|
| | Local | LRT | |
| $K^+$/OC | 0.4 | 0.2 | |
| $K^+$/EC | 0.9 | 0.4 | |
| $K^+$/Zn | 3.7 | 2.7 | |
| $K^+$/$NO_3^-$ | 3.6 | 0.1 | |
| | LRT episodes | | |
| | Secondary sulfate | industrial | Resuspended fine dust |
| $(Ca^{2+}+Mg^{2+})$/$Na^+$ | 5.7 | 1.6 | null |
| $NO_3^-$/$SO_4^{2-}$ | null | 0.9 | 0.1 |