# Peer review of "Seasonal variation and chemical characterization of PM2.5 in northwestern Philippines"

_Atmospheric Chemistry and Physics, 2017_

## Referee Comment (RC1) · Anonymous Referee #1 · 8 Dec 2017

This paper describes the seasonal change of PM2.5 characteristics on the basis of the sampling data taken at the northern part of the Philippines. The chemical component analysis coupled with the back trajectory study has revealed that the fine aerosols are composed of natural origins, local emissions, and long range transport effects. As a whole, the paper conveys useful insight into the air quality analysis in the quoted region. In order to improve the quality of the paper, the reviewer recommends the authors to consider the following issues.

(major) p.6 Please give a brief explanation why the OC/EC ratio below 2 indicates the dominance of primary aerosol. p.6 "daytime sea breeze pushes back these polluted air masses inland": is there any observational evidence or supporting data for this situation? p.8 "heavy metal components come from several different sources": what

are the most plausible sources? p.8 "the ratio of NH4+/SO4 2-": isn't is necessary to consider the charge balance in this case? If so, the ratio between 2(NH4 +) and (SO4 2-) must be considered instead? p.8 "the minimum Q value": a brief explanation of the Q value would be of help. p.9 A brief explanation if the "enrichment factor" will be of help. p.10 The paragraph describing figure 12 should be moved to the text, not conclusion. p.17 If possible, it would be better to move the panel indications (a)-(d) to just above each panel, not below. Moreover showing the season nearby the panel as (a) Summer, for example, will be effective for seeing the differences in the four seasons. (The same applies to other figures.) p.17 Fig. 2: the unit mm should be shown at the side of the color bar. What exactly was the accumulation time for the "accumulated rainfall"? p.18 Fig.3: at the vertical axis, the unit ug/m3 should be shown with parentheses. In fig. 4 the vertical axis should show the quantity, not only the unit. p.19 Fig. 5 and p.21 Fig. 8: it would be better to employ different symbols (such as open circle, filled circle, etc.) to indicate different seasons. Also, the meaning of each line must be explained in the caption. p.20 Fig. 6: the panels (a), (b), ...should be mentioned in the caption. The same for Fig. 7. p.22, Fig. 8: The unit (ueq m-3) should be shown with parentheses. p.23 Fig. 9: panels should appear with (a) - (f). The same for Fig. 10. p.25 Fig. 11 The axes should be with the quantity and unit, not just (modeled) and (measured). p.26 Fig. 12: in the caption, the difference in four seasons should be explained explicitly. By using different symbols for different seasons, the figure would be more directly understandable.

(minor) p.1 "The seasonal and chemical characteristic of ... was" -> The seasonal and chemical characteristics of ... were p.1 The values of 21.59 and 8.44 should be 21.6 and 8.4 ug m-3, respectively. p.1 "USEPA PMF" should be spelled out. (In p.4, it is spelled as "US EPA".) p.1, p.2 "Long Range Transport (LRT)" should be "long range transport (LRT)"? Please check the policy of the journal. p.1 "LRT of industrial emission ... have" should be "The LRTs of industrial emission, ... have" p.1 "Aerosols are known ... but also on its effects on ...": "its" should be "their" p.1 "Rapid industrialization ... has led to": "has" should be "have". p.1 "High emissions from ... is transported

...": "is" should be "are". p.2 "factors like" should be "factors such as". p.2 "Its effects cover large regions of SEA.": this part should be connected to the previous sentence, for instance as ", the effects of which cover ...". p.2 "The life cycle of these aerosols and its impacts on ...": what are "these aerosols"? "its" should be "their"? p.2 "the main source of regional stratospheric air.": is this part correctly describing the exact situation? p.2 "Leading to observed elevated levels of ...": an incomplete sentence. p.3 "atop a 3 storey building": "storey" should be "story". p.3 "Except for the summer sampling period": an incomplete sentence. p.4 "One for" should be "one for". p.4 "without heating the filter" should be "without heating. The filter". p.4 "FNL final reanalysis": FNL should be explained. p.4 "disaggregated": is this a proper wording? p.4 "in this study" should be "in the present study". p.5 "from the SCS making" should be "from the SCS, making". p.5 "And in fig. 2d": please avoid starting a sentence with "And". p.5 "early 2016 are" should be "early 2016 were". p.5 "we expect its sources to ....": the meaning of "its" is not clear. p.5 "to be significant influence by" should be "to be significantly influenced by". p.5 "0.67 $\pm$ 0.3" should be "0.67 $\pm$ 0.30". p.5 "Measured EC likely" should be "Measured EC is likely". p.7 "used as tracer": "used as a tracer". p.7 "shown table 2": "shown in table 2". p.7 Please italicize variables r (correlation coefficient) and p (confidence interval). p.7 "in the proceeding section": "in the following section". p.7 "The seasonal concentrations of which are": an incomplete sentence. p.7 "Seasonal variation of heavy metals ... are": "Seasonal variations of heavy metals ... are". p.7 "Ambient concentration of ... depend on distance": "Ambient concentrations of ... depend on the distance". p.8 "the ratio ... are also found to be": "the ratio .... is also found to be". p.9 "the wind back trajectory analysis discussed": "the wind back trajectory analysis discussed above". p.9 "strong associations with the heavy metals ...": the heavy metals are not quoted in table 4 discussed here. p.9 "mass concentration. Providing": "mass concentration, providing" p.9 "This study describes": "This study has described". p.10 "respectively. Both": "respectively, both". p.10 "air masses originating from East Asia moves": "moves" should be "move" p.10 "when lowest": "when the lowest". p.16 The reference Zhu et al., 2017 should be listed in the reference section, not

below table 1.
* * *
[Figure]

---

## Referee Comment (RC2) · Anonymous Referee #2 · 21 Dec 2017

This manuscript presents the first seasonal analysis of the fine particulate matter and its components in Burgos, and also discussed the source attribution using the PMF model. Though for each season, the study only had 7-14 days of sample, this study provides a peak for the magnitude and seasonal distribution of the PM2.5 ÂňÂňdistribution in this area. In my opinion, this paper is written poorly. Sentences were sometimes not complete or, too long with comma only. The authors should spend time and effort to revisit their draft and improve the writing. Some specific comments can be seen below.

Major comments: 1.In the abstract, add the standard deviation for the the peak and low concentration of the PM2.5. Also, keep consistent for the valid digit used in the paper. For example, in the abstract, the authors listed the highest PM2.5 of 21.59, but in the section 3.2, it listed 21.6.

2.The discussions of the transition of the monsoons under section 3.2 are not very appropriate, or even very redundant. I didn't see any connections between these few paragraphs with other contents. Suggest the authors remove these discussions, or put them together with the source attribution under section 3.4 to help explain the sources of PM2.5 over this area.

3.In section 2.1, the authors discussed that the observation period during summer for this study was a "monsoon break", which makes all sampling periods non-rainy days. This makes me wonder how will that affect the seasonal distribution of the aerosols over this area, and how the authors' conclusion "peak in spring and low in fall" will stand out. Precipitations should have significant impact on aerosol. So please explain or add to the discussions.

4.Explain the enrichment factor.

Minor comments 1.Reorder all the figures. The figure number start with 1 instead of 11.

2.Pg 1 line 2: This study only has 7 days of observation during summer. So please clarify.

3.Pg 1 line 16: change "but also on its effects" to "but also for their effects"

4.Pg 1 line 17: cite the latest IPCC 2013 report.

5.Pg 1 line 20: "is transported" to "are transported"

6.Pg 2 line 10: incomplete sentence.

7.Pg 2 line 33-35: In this paragraph, the authors started to discuss the LRT on the influence of the aerosol in the county. Then they switched to discuss that this region is also source of biomass burning emissions. The authors should make a new paragraph discuss on the differences between the LRT and regional sources on local aerosol concentration. No need to capitalize Organic Carbon.

[Figure]

8.Pg 4 line 5-9: rewrite these sentences.

9.Pg 5 line 24-26: use the seasonal mean plus STD to discuss the seasonal differences since the values showed in the manuscript are from daily values which are meaningless.

10.Pg 6 line 3: put the "the bold dashed line ..." into figure 5 instead of the main contents.

11.Pg 7 line 4-5: rewrite the sentence.

12.Pg 9 line 25: delete the last half sentence or rewrite as a whole.

13.Pg 10: in the conclusion part, add the discussions of the seasonality of the total PM2.5, which is the main points of this study.

14.Pg 10 line 23-27: consider to move this paragraph into the results.

15.Pg 18, Figure 4: I suggest the authors make a similar plot as Fig. 3 for both OC and EC, by doing that both the temporal characteristic of OC and BC, and also their ratios can be clearly seen.

16.Pg 18, Figure 5: choose different markers for the OC/EC ratio plots.

17.Pg 21, Figure 8 (c): change Ca to "Ca2+"

---

## Author Comment (AC1) · 8 Jan 2018

This paper describes the seasonal change of PM2.5 characteristics on the basis of the sampling data taken at the northern part of the Philippines. The chemical component analysis coupled with the back trajectory study has revealed that the fine aerosols are composed of natural origins, local emissions, and long range transport effects. As a whole, the paper conveys useful insight into the air quality analysis in the quoted region. In order to improve the quality of the paper, the reviewer recommends the authors to consider the following issues. Response. We would like to thank you for the constructive comments and we really appreciate the reviewer's patience especially for the minor comments. (major)

[Figure]

p.6 Please give a brief explanation why the OC/EC ratio below 2 indicates the dominance of primary aerosol. Response. EC mostly comes from primary combustion sources while OC can come from both primary and secondary (gas-to-particle conversion). For this, the ratio of the 2 carbonaceous components is usually used in source apportionment of aerosols. The ratio threshold value of 2 was set by Chow et al. 1996. Added in section 3.2 "EC and OC are good tracers for fossil fuel combustion and biomass burning, respectively. EC is only of primary origin while OC may be emitted directly or form by gas-to-particle conversion in the atmosphere\citep{jones2005}. Accordingly, OC/EC ratio is usually used in source apportionment of carbonaceous aerosols \citep{pio2011}."

references added:

Jones, A.M. and Harrison, R.M.: Interpretation of particulate elemental and organic carbon concentrations at rural, urban and kerbside sites. Atmospheric Environment, 39(37), pp.7114-7126, 2005.

Pio, C., Cerqueira, M., Harrison, R.M., Nunes, T., Mirante, F., Alves, C., Oliveira, C., de la Campa, A.S., Artíñano, B. and Matos, M.: OC/EC ratio observations in Europe: Re-thinking the approach for apportionment between primary and secondary organic carbon. Atmospheric Environment, 45(34), pp.6121-6132, 2011.

p.6 "daytime sea breeze pushes back these polluted air masses inland": is there any observational evidence or supporting data for this situation? Response. No, there is no observational data supporting this, just from model. However, one of the co-authors found similar situation along the SW Taiwan coasts (Tsai et al.: Effects of sea-land breezes on the spatial and temporal distribution of gaseous air pollutants at the coastal region of southern Taiwan. J. Environ. Eng. Manag, 18, pp.387-396, 2008). We think this can be a starting point on studies on coastal effects. Does the reviewer suggests that we remove this part?

p.8 "heavy metal components come from several different sources": what are the most

plausible sources? Response. Our hypothesis is that local sources has minimal contribution (i.e., smoke coming from exhausts of buses and trucks from adjacent road). Heavy metal components have high concentration in winter and spring time, thus pointing to LRT. This is the topic of another manuscript in preparation where we hypothesize that emissions from Liaoning province in northeast China as a significant source of transboundary pollution (including heavy/toxic metals) in northern Philippines. Liaoning is the largest provincial economy in northern China with industries such as oil refinery, source of petroleum and natural gas, mining, metal refining, various chemical industries, etc.

p.8 "the ratio of NH4+/SO4 2-": isn't is necessary to consider the charge balance in this case? If so, the ratio between 2(NH4 +) and (SO4 2-) must be considered instead? Response: We use equivalent concentrations in calculating the ratio, hence, the number of moles NH4+ versus SO4-2 is already considered.

p.8 "the minimum Q value": a brief explanation of the Q value would be of help. Response: The Q value is the object function of the PMF algorithm. The PMF is a weighted least squares model: weighted based on the matrix containing a known uncertainty and known concentration of the chemical species of interest. In the PMF the model algorithm must fit that of measured values (using the context of mass balance). Anything in excess will be assigned as the residual (hence, residuals are the difference between the model and measurement fittings). Dividing the residuals to the uncertainty values then normalizes the residuals (normalization function). The sum of the square of this normalization function is the object function; Q. Hence, identifying the least (minimum) Q is the first step in finding the right fit of the PMF algorithm. As the PMF method has become popular, we deem it unnecessary to further detail the method. But we would comply if the author would suggest us to do so.

p.9 A brief explanation if the "enrichment factor" will be of help. Response: Enrichment factor is a method of characterizing the chemical composition of a metallic element determining the abundance of elements by using a reference. Enrichment factor is the

is an approach established by Taylor (1964) to characterize the chemical composition of airborne particulate matter (APM) by relating the concentration of an element to that of a crustal element in the air, normalized to the ratio of the element in the average continental crust (Farooq et al., 2012). The method uses reference elements, usually those that are stable in the soil, and are least influenced by vertical shear, and/or anthropogenically altered (Ackermann, 2008).

We've added the following in the manuscript: "Analysis of the enrichment factor (Taylor, 1964, Hernández-Mena et al., 2011; Lomboy et al., 2015; Rushdi et al., 2013) is done to further characterize the composition and associations of the chemical components of PM2.5. The analysis employs relating the concentration of PM2.5 components that are known anthropogenic to those that are found stable in the crust, or those that are naturally found in the local atmosphere. "

References: Ackermann, F. (2008). A procedure for correcting the grain size effect in heavy metal analyses of estuarine and coastal sediments A PROCEDURE FOR COR-RECTING THE GRAIN SIZE EFFECT IN HEAVY METAL ANALYSES OF, (September 2013), 37–41. Farooq, H., Ahmad, M. R., Jamil, Y., Ahmad, M. R., Khan, M. A. A., Mahmood, T., . . . Khan, S. A. (2012). Lead Pollution Measurement Motorway in Punjab , Pakistan Along National Highway and. Journal of Basic and Applied Sciences, 8, 463–467. http://doi.org/10.6000/1927-5129.2012.08.02.34 Hernández-Mena, L., Murillo-Tovar, M., Ramírez-Muñíz, M., Colunga-Urbina, E., De La Garza-Rodríguez, I., & Saldarriaga-Noreña, H. (2011). Enrichment factor and profiles of elemental composition of PM 2.5 in the city of Guadalajara, Mexico. Bulletin of Environmental Contamination and Toxicology, 87(5), 545–549. http://doi.org/10.1007/s00128-011-0369-x Lomboy, M. F. T. C., Quirit, L. L., Molina, V. B., Dalmacion, G. V., Schwartz, J. D., Suh, H. H., & Baja, E. S. (2015). Characterization of particulate matter 2.5 in an urban tertiary care hospital in the Philippines. Building and Environment, 92, 432–439. http://doi.org/10.1016/j.buildenv.2015.05.018 Rushdi, A. I., Al-Mutlaq, K. F., Al-Otaibi, M., El-Mubarak, A. H., & Simoneit, B. R. T. (2013). Air quality and elemental enrichment factors of aerosol particulate matter in Riyadh City, Saudi Arabia. Arabian Journal of Geosciences, 6(2), 585–599. http://doi.org/10.1007/s12517-011-0357-9 Taylor, S. R. (1964). Abundance of chemical elements in the continental crust: a new table. Geochimica et Cosmochimica Acta, 28(8), 1273–1285. http://doi.org/10.1016/0016-7037(64)90129-2

p.10 The paragraph describing figure 12 should be moved to the text, not conclusion. Response: That discussion is moved to section 3.2.4

p.17 If possible, it would be better to move the panel indications (a)-(d) to just above each panel, not below. Moreover showing the season nearby the panel as (a) Summer, for example, will be effective for seeing the differences in the four seasons. (The same applies to other figures.) Response: Edited as suggested

p.17 Fig. 2: the unit mm should be shown at the side of the color bar. What exactly was the accumulation time for the "accumulated rainfall"? Response: Edited as suggested. Accumulation for the whole sampling period for each season. This is now reflected on the figure caption. caption{Wind (arrows), accumulated rainfall for each sampling period (shading, in \unit{mm}) and wind back trajectory (red line) during a) summer, b) fall, c) winter and d) spring sampling season. The gray scale is white for a value of 0 and goes to black for a value of 250 in increments of 50\unit{mm}. }

p.18 Fig.3: at the vertical axis, the unit ug/m3 should be shown with parentheses. In fig. 4 the vertical axis should show the quantity, not only the unit. Response: Edited

p.19 Fig. 5 and p.21 Fig. 8: it would be better to employ different symbols (such as open circle, filled circle, etc.) to indicate different seasons. Also, the meaning of each line must be explained in the caption. Response: Edited as suggested

p.20 Fig. 6: the panels (a), (b), ...should be mentioned in the caption. The same for Fig. 7. Response: Edited as suggested

p.22, Fig. 8: The unit (ueq m-3) should be shown with parentheses. Response: Edited

as suggested

p.23 Fig. 9: panels should appear with (a) - (f). The same for Fig. 10. Response: Edited as suggested

p.25 Fig. 11 The axes should be with the quantity and unit, not just (modeled) and (measured). Response: Edited to include quantity and unit

p.26 Fig. 12: in the caption, the difference in four seasons should be explained explicitly. By using different symbols for different seasons, the figure would be more directly understandable. Response: Figure description edited to explicitly include seasonal description

(minor) p.1 "The seasonal and chemical characteristic of ... was" -> The seasonal and chemical characteristics of ... were Response: Edited as suggested p.1 The values of 21.59 and 8.44 should be 21.6 and 8.4 ug m-3, respectively. Response: Edited as suggested p.1 "USEPA PMF" should be spelled out. (In p.4, it is spelled as "US EPA".) Response: Edited throughout the manuscript as suggested p.1, p.2 "Long Range Transport (LRT)" should be "long range transport (LRT)"? Please check the policy of the journal. Response: Edited to lower case following published articles. p.1 "LRT of industrial emission ... have" should be "The LRTs of industrial emission, ... have" Response: Edited as suggested p.1 "Aerosols are known ... but also on its effects on ...": "its" should be "their" Response: Edited p.1 "Rapid industrialization ... has led to": "has" should be "have". Response: Edited p.1 "High emissions from ... is transported ...": "is" should be "are". Response: Edited p.2 "factors like" should be "factors such as". Response: Edited p.2 "Its effects cover large regions of SEA.": this part should be connected to the previous sentence, for instance as ", the effects of which cover ...". Response: Edited p.2 "The life cycle of these aerosols and its impacts on ...": what are "these aerosols"? "its" should be "their"? Response: Edited p.2 "the main source of regional stratospheric air.": is this part correctly describing the exact situation? Response: Edited stratospheric air to troposphere-to-stratospheric transported

air p.2 "Leading to observed elevated levels of ...": an incomplete sentence. Response: Edited p.3 "atop a 3 storey building": "storey" should be "story". Response: Edited p.3 "Except for the summer sampling period": an incomplete sentence. Response: Edited p.4 "One for" should be "one for". Response: Edited p.4 "without heating the filter" should be "without heating. The filter". Response: Edited p.4 "FNL final reanalysis": FNL should be explained. Response: Edited FNL to National Centers for Environmental Prediction or NCEP FNL (final) global reanalysis p.4 "disaggregated": is this a proper wording? Response: This is the terminology often used in describing PMF p.4 "in this study" should be "in the present study". Response: Edited throughout the manuscript as suggested p.5 "from the SCS making" should be "from the SCS, making". Response: Edited p.5 "And in fig. 2d": please avoid starting a sentence with "And". Response: Noted with thanks p.5 "early 2016 are" should be "early 2016 were". Response: Edited p.5 "we expect its sources to ....": the meaning of "its" is not clear. Response: Edited p.5 "to be significant influence by" should be "to be significantly influenced by". Response: Edited p.5 "0.67 ± 0.3" should be "0.67 ± 0.30". Response: Edited p.5 "Measured EC likely" should be "Measured EC is likely". Response: Edited p.7 "used as tracer": "used as a tracer". Response: Edited p.7 "shown table 2": "shown in table 2". Response: Edited p.7 Please italicize variables r (correlation coefficient) and p (confidence interval). Response: Edited p.7 "in the proceeding section": "in the following section". Response: Edited p.7 "The seasonal concentrations of which are": an incomplete sentence. RResponse: Edited p.7 "Seasonal variation of heavy metals ... are": "Seasonal variations of heavy metals ... are". Response: Edited p.7 "Ambient concentration of ... depend on distance": "Ambient concentrations of ... depend on the distance". Response: Edited p.8 "the ratio ... are also found to be": "the ratio .... is also found to be". Response: Edited p.9 "the wind back trajectory analysis discussed": "the wind back trajectory analysis discussed above". Response: Edited p.9 "strong associations with the heavy metals ...": the heavy metals are not quoted in table 4 discussed here. Response: We are referring to the source factor profiles of fig. 9, where heavy metal components are comparatively high p.9 "mass concentration. Providing":

"mass concentration, providing" Response: Edited p.9 "This study describes": "This study has described". Response: Edited p.10 "respectively. Both": "respectively, both". Response: Edited p.10 "air masses originating from East Asia moves": "moves" should be "move" Response: Edited p.10 "when lowest": "when the lowest". Response: Edited p.16 The reference Zhu et al., 2017 should be listed in the reference section, not below table 1. Response: This seems to be due to the (automatic) rendering of latex.

---

## Author Comment (AC2) · 19 Jan 2018

This manuscript presents the first seasonal analysis of the fine particulate matter and its components in Burgos, and also discussed the source attribution using the PMF model. Though for each season, the study only had 7-14 days of sample, this study provides a peak for the magnitude and seasonal distribution of the PM2.5 ÂnÂËĞ ndistribu- ËĞ tion in this area. In my opinion, this paper is written poorly. Sentences were sometimes not complete or, too long with comma only. The authors should spend time and effort to revisit their draft and improve the writing. Some specific comments can be seen below. Response. We appreciate and would like to thank the reviewer for the constructive comments. We will do our best in editing this manuscript to improve the writing. As to the 7-14 day sampling to represent a season, we think that the

sampling days/periods were able to capture the climatological characteristics of each of the monsoon regimes (southwest, northeast and transition). Moreover, another year of sampling was done (not included in this present study) in slightly different months but in the same seasonal partition (i.e. mam, jja, son and djf), data shows similar seasonal wind and concentration variability. Leading us to conclude that the sampling we did for this study is representative of the seasons.

1.In the abstract, add the standard deviation for the the peak and low concentration of the PM2.5. Also, keep consistent for the valid digit used in the paper. For example, in the abstract, the authors listed the highest PM2.5 of 21.59, but in the section 3.2, it listed 21.6. Response. Manuscript edited to keep decimal places consistent and include standard deviation in abstract.

2.The discussions of the transition of the monsoons under section 3.2 are not very appropriate, or even very redundant. I didn't see any connections between these few paragraphs with other contents. Suggest the authors remove these discussions, or put them together with the source attribution under section 3.4 to help explain the sources of PM2.5 over this area. Response. Since the Philippines is in the tropics, the 4 boreal seasons do not necessarily describe the local seasons. Moreover, local seasonal terminologies in the Philippines are at times contradictory to common/normal climate descriptions. For instance, the dryest months in the Philippines are from March to May, and this is locally reffered to as "summer". With these inconsistencies (especially for local readers) in mind, we think it is best to retain this brief discussion of boreal season – monsoon relation. Does the reviewer suggests this section be moved to 3.4 or be incorporated in 3.2?

3.In section 2.1, the authors discussed that the observation period during summer for this study was a "monsoon break", which makes all sampling periods non-rainy days. This makes me wonder how will that affect the seasonal distribution of the aerosols over this area, and how the authors' conclusion "peak in spring and low in fall" will stand out. Precipitations should have significant impact on aerosol. So please explain or add to

the discussions. Response: Monsoon break here is meant to describe a reduction of convective activity in the region. The summer season of JJA is the time when tropical cyclones in the northwest Pacific is most active. At this time, significant rainfall along the northwest coast of the Philippines is induced when a tropical cyclone is present to the east of the Philippines (Bagtasa 2017). During the summer sampling period, a typhoon just passed by the region, and there was no synoptic scale disturbance present in the northwest Pacific region. For other seasons, on the other hand, the northwest region of the Philippines has a distinctly dry climate (Coronas 1912, page 3 line 20), the sampling days other than summer season were characteristic of dry season. In addition, our sampling protocol calls for the temporary suspension of sampling whenever there is rain, this is to prevent abnormally low aerosol concentrations that may pull down the mean 24h concentration values. Whenever sampling is suspended, we do the necessary correction on the number of sampling hours in calculating concentration. It just so happened that all sampling days in this study had non-rainy days and there was no need to suspend any sampling.

4.Explain the enrichment factor. Response: Enrichment factor is a method of characterizing the chemical composition of a metallic element determining the abundance of elements by using a reference. Enrichment factor is the is an approach established by Taylor (1964) to characterize the chemical composition of airborne particulate matter (APM) by relating the concentration of an element to that of a crustal element in the air, normalized to the ratio of the element in the average continental crust (Farooq et al., 2012). The method uses reference elements, usually those that are stable in the soil, and are least influenced by vertical shear, and/or anthropogenically altered (Ackermann, 2008). We've added the following in the manuscript: "Analysis of the enrichment factor (Taylor, 1964, Hernández-Mena et al., 2011; Lomboy et al., 2015; Rushdi et al., 2013) is done to further characterize the composition and associations of the chemical components of PM2.5. The analysis employs relating the concentration of PM2.5 components that are known anthropogenic to those that are found stable in the crust, or those that are naturally found in the local atmosphere. " References: Ackermann, F. (2008). A procedure for correcting the grain size effect in heavy metal analyses of estuarine and coastal sediments A PROCEDURE FOR CORRECTING THE GRAIN SIZE EFFECT IN HEAVY METAL ANALYSES OF, (September 2013), 37–41. Farooq, H., Ahmad, M. R., Jamil, Y., Ahmad, M. R., Khan, M. A. A., Mahmood, T., . . . Khan, S. A. (2012). Lead Pollution Measurement Motorway in Punjab , Pakistan Along National Highway and. Journal of Basic and Applied Sciences, 8, 463–467. http://doi.org/10.6000/1927-5129.2012.08.02.34 Hernández-Mena, L., Murillo-Tovar, M., Ramírez-Muñíz, M., Colunga-Urbina, E., De La Garza-Rodríguez, I., & Saldarriaga-Noreña, H. (2011). Enrichment factor and profiles of elemental composition of PM 2.5 in the city of Guadalajara, Mexico. Bulletin of Environmental Contamination and Toxicology, 87(5), 545–549. http://doi.org/10.1007/s00128-011-0369-x Lomboy, M. F. T. C., Quirit, L. L., Molina, V. B., Dalmacion, G. V., Schwartz, J. D., Suh, H. H., & Baja, E. S. (2015). Characterization of particulate matter 2.5 in an urban tertiary care hospital in the Philippines. Building and Environment, 92, 432–439. http://doi.org/10.1016/j.buildenv.2015.05.018 Rushdi, A. I., Al-Mutlaq, K. F., Al-Otaibi, M., El-Mubarak, A. H., & Simoneit, B. R. T. (2013). Air quality and elemental enrichment factors of aerosol particulate matter in Riyadh City, Saudi Arabia. Arabian Journal of Geosciences, 6(2), 585–599. http://doi.org/10.1007/s12517-011-0357-9 Taylor, S. R. (1964). Abundance of chemical elements in the continental crust: a new table. Geochimica et Cosmochimica Acta, 28(8), 1273–1285. http://doi.org/10.1016/0016-7037(64)90129-2

(minor) 1.Reorder all the figures. The figure number start with 1 instead of 11. Response: Edited. We apologize for this as it was a problem with latex typesetting.

2.Pg 1 line 2: This study only has 7 days of observation during summer. So please clarify. Response: Edited. "Each 24H sample ...for two weeks every season" to "Each 24H sample ...for four seasons from 2015 – 2016". The detailed discussion of sampling period is then found in sec. 2.2.

3.Pg 1 line 16: change "but also on its effects" to "but also for their effects" Response:

Edited.

4.Pg 1 line 17: cite the latest IPCC 2013 report. Response: citation updated.

5.Pg 1 line 20: "is transported" to "are transported" Response: Edited.

6.Pg 2 line 10: incomplete sentence. Response: Edited. From ". Its effects cover large regions of SEA." to "..., the effects of which cover large regions of SEA."

7.Pg 2 line 33-35: In this paragraph, the authors started to discuss the LRT on the influence of the aerosol in the county. Then they switched to discuss that this region is also source of biomass burning emissions. The authors should make a new paragraph discuss on the differences between the LRT and regional sources on local aerosol concentration. No need to capitalize Organic Carbon. Response: Edited. Study about the Philippines as source of levoglucosan moved to 2nd paragraph.

8.Pg 4 line 5-9: rewrite these sentences. Response: Edited. From "..., the filter" to ". The filter..."

9.Pg 5 line 24-26: use the seasonal mean plus STD to discuss the seasonal differences since the values showed in the manuscript are from daily values which are meaningless. Response: Edited. Reference to daily values were removed.

10.Pg 6 line 3: put the "the bold dashed line . . ." into figure 5 instead of the main contents. Response: Edited.

11.Pg 7 line 4-5: rewrite the sentence. Response: Edited. From "Seasonal mean and standard deviation of PM2.5 and some water soluble ionic components are shown in fig. 6. Figures 6b, 6c and 6d show $NO_{-3}$ , $SO2_{-}4$ and $NH_{+}4$ , respectively" to "Figure 6 shows the seasonal mean and standard deviation of PM2.5 and some water soluble ionic components. It is apparent that $NO_{-3}$, $SO2_{-}4$ and $NH_{+}4$ shown in fig. 6b, 6c and 6d , respectively," (appropriate formating is applied in manuscript)

12.Pg 9 line 25: delete the last half sentence or rewrite as a whole. Response: Edited.

From "...mass concentration. Providing" to "mass concentration, providing"

13.Pg 10: in the conclusion part, add the discussions of the seasonality of the total PM2.5, which is the main points of this study.

14.Pg 10 line 23-27: consider to move this paragraph into the results. Response: Edited. Moved as sec 3.2.4

15.Pg 18, Figure 4: I suggest the authors make a similar plot as Fig. 3 for both OC and EC, by doing that both the temporal characteristic of OC and BC, and also their ratios can be clearly seen. Response: Figure edited to include daily OC/EC ratio.

16.Pg 18, Figure 5: choose different markers for the OC/EC ratio plots. Response: Figure marker modified, same for fig. 8.

17.Pg 21, Figure 8 (c): change Ca to "Ca2+" Response: Corrected.

---

## Author Response (AR2)

Author's response

1. Page 1 line 6: change "sea salt is" to "sea salt was". Please read the paper carefully and make revisions for other places.

Response: Edited to conform to suggestion.

2. For figs 5 and fig. 9, the colors for the outlines and markers are not consistent.

Response: fig. 5 color adjusted for consistency.